# Monolithic scalable compliant mechanisms

**Jared R. Hunter**[1,☉], **Bethany Parkinson**[1,☉]*, **Jacob L. Sheffield**[1], **Mark B. Rober**[2], **Brian D. Jensen**[1], **Spencer P. Magleby**[1], **Nathan S. Usevitch**[1], **Larry L. Howell**[1]

**1** Department of Mechanical Engineering, Brigham Young University, Provo, Utah, United States of America, **2** CrunchLabs LLC, Sunnyvale, California, United States of America

☉ These authors contributed equally to this work.
* bethanyparkinson@gmail.com

## Abstract

Scaling a physical device's geometry results in mechanical properties changing in various ways (e.g. the cubed-squared law states that for a scaling factor $C$, mass scales with $C^3$ and surface area with $C^2$). These scaling effects can result in a device's inconsistent and unplanned mechanical behavior when varying its fabricated size, thereby necessitating unique designs at different scales. We show that for displacement-driven compliant mechanisms, mechanical stress is uniquely invariant with scale. This effect is described theoretically, verified through computer models and physical testing, and is demonstrated in three examples: a parallel-guiding mechanism, a projectile launcher, and a deployable chair. This enhanced understanding of stress invariance provides innovative insight into the way devices can be designed for systems that operate across different scales.

## Introduction

Compliant mechanisms transfer motion, force, or energy through the deflection of flexible members rather than through traditional mechanism components such as hinges or bearings. Their potential advantages include more precise motion, decreased part count, simpler manufacturing, decreased cost, lower mass, reduced friction, and increased portability [1,2]. Compliant mechanisms often develop in nature [3]. For example, an elephant's trunk does not have traditional hinges or joints, but is able to perform complex functions through actuation of flexible systems. Similarly, leaves and tree branches benefit from their ability to flex under the force of wind without fracturing [4], plants make use of compliance at the cell level to achieve osmosis [5], and the wings of many animals, including birds, bees, and ladybugs [6,7], benefit from flexibility to improve aerodynamic performance and to maximize energy storage [8,9]. One additional potential advantage of compliant mechanisms is their ability to be made of a single piece. This monolithic potential of compliant mechanisms results in recognized advantages and challenges (e.g. difficult to analyze and design, material fatigue limits).

reproduction in any medium, provided the original author and source are credited.

**Data availability statement:** All relevant data are within the manuscript and its Supporting information files.

**Funding:** This work was supported through funds awarded to BDJ by Castle Grayskull, LLC, founded by Mark Rober. The funder did not participate in data collection and analysis, but did provide support in the form of salaries for authors JH, BP, and JS and was involved in the study design, decision to publish, and preparation of the manuscript. The specific roles of these authors are articulated in the 'author contributions' section.

**Competing interests:** I have read the journal's policy and the authors of this manuscript have the following competing interests: This work was supported through funds awarded to BYU by Castle Grayskull, LLC, owned by Mark Rober. These funds supported authors JH, BP, and JS in the form of salaries. Mark Rober is also one of the authors of this paper and was involved in the study design, decision to publish, and preparation of the manuscript. A patent application is pending on the planar spring that is integral to the blaster case study. A patent has been issued on the LET joints that are part of the chair case study. Another of the mechanisms presented in this paper as a case study was previously described in U.S. Patent Publication No. US 2025/0102033 A1, entitled "SCALEABLE FLAT IN-PLANE-MOTION MECHANICAL SPRING." This does not alter our adherence to PLOS ONE policies on sharing data and materials.

One trait of compliant mechanisms that has not been previously articulated is their ability to satisfy conditions that enable scaling designs to different sizes without scaling their maximum stress. This scalability trait exhibited by displacement-driven compliant mechanisms shows promise as a means to enable the design of mechanical devices that can maintain relative geometry and predictable behavior when scaled to different sizes. This understanding facilitates the creation of new systems that were not previously possible.

We first illustrate this discovery using a straightforward compliant mechanism: the parallel-guiding mechanism illustrated in Fig 1A. We then demonstrate the concept and indicate its implications through more complex compliant mechanisms: a single-piece compliant projectile launcher (see Fig 1B) and a deployable compliant structure, a chair folded from a single sheet of material (see Fig 1C).

Scalability of compliant mechanisms are of particular interest for Micro-Electro-Mechanical Systems (MEMS) applications [10,11] since rigid-body mechanisms often contain components which are difficult to manufacture or assemble at the microscale. Medical devices on the meso or microscale are also a motivating factor [12–16]. In the other direction, applications of compliant mechanisms in space may require scaling to the macroscale [17]. Previous work has investigated how scaling affects resultant deflection, angle of linkage deflection, and strain [10]; natural frequency [18]; and stiffness [19]. Through understanding principles of scalability, designers can design and prototype initially for the macroscale, where prototyping is simpler and may be performed more quickly (e.g. through use of 3D printers or traditional prototyping methods), before easily scaling up or down their design for the final iterations. Since manufacturing at extreme scales is often more complex and expensive [20,21], understanding scaling principles has the potential to improve the design process for the applications of interest as listed above.

We build on these previous works by presenting a comprehensive set of conditions and properties for displacement-driven compliant mechanisms and how they each scale with geometry. These properties are applicable to displacement-driven compliant mechanisms in general. We demonstrate the models for the properties through a variety of compliant mechanisms. Compliant mechanism synthesis may be performed through a variety of methods, traditionally including the pseudo-rigid body model [1,22] and topology optimization [23–26]. More recent methods include behavior modeling through neural networks [27] and single-point synthesis [28]. By investigating these models for scaling properties of compliant mechanisms, we add another tool to the compliant mechanism synthesis toolbox and thus provide designers with the ability to more easily create highly tailorable and scalable mechanisms.

## Methods

The effects of scaling on different mechanical properties were recognized when doing an atypical project that extended compliant mechanisms theory to a broad audience in a STEM video series where the public is introduced to science and engineering principles in entertaining ways [29]. Compliant mechanisms were used to create

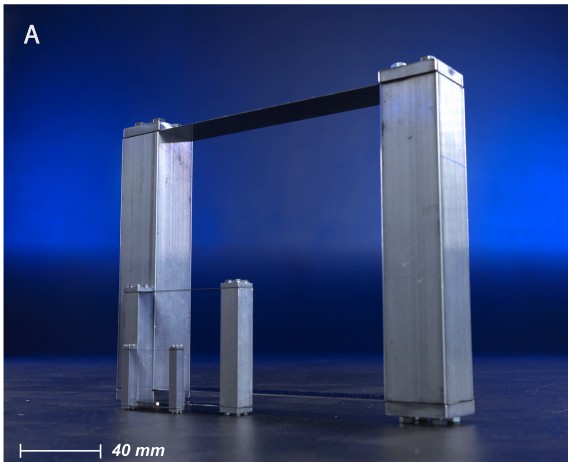

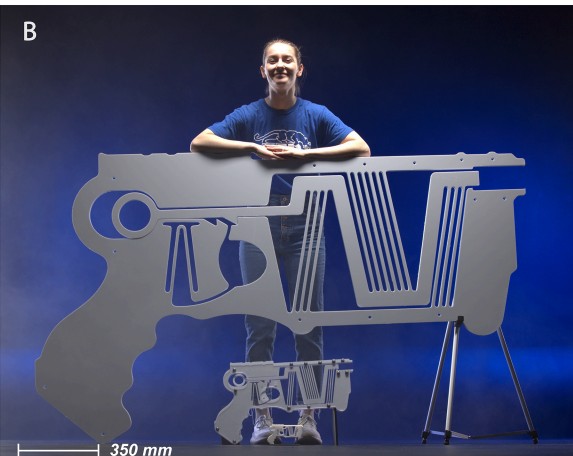

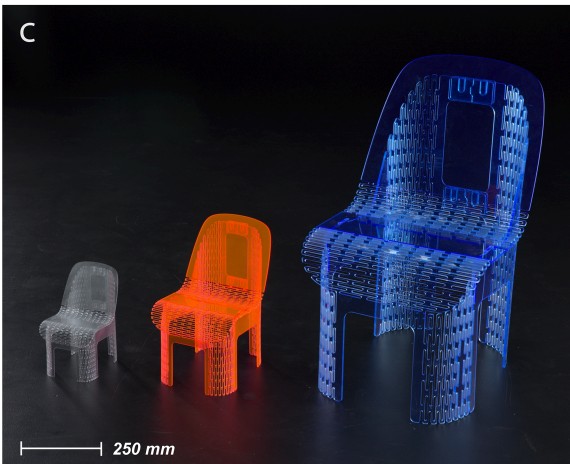

**Fig 1**. **Displacement-driven compliant mechanisms exhibit a unique trait of scalability that unlocks potential for novel mechanical designs to operate at various scales.** (A) Three sizes of a parallel-guided mechanism. (B) Three sizes of a one-piece, fully compliant projectile launcher, with a person for scale. (C) Three sizes of a monolithic chair. The individual in this figure has given written informed consent (as outlined in PLOS consent form) to have their image appear in this paper.

the world's smallest dart launcher [30], and, for the video to be effective, the device needed to have the same appearance and relative behavior at several scales. This is a challenge because different mechanical properties scale at different rates. This unusual (even unorthodox) application revealed conditions that enabled the system to be scaled while the maximum stress was invariant across scales. Those conditions can be generalized in a way that benefit a broad range of applications.

In this section, we derive equations describing stress scalability behavior of a displacement-driven compliant mechanism: the parallel-guided mechanism. The design of a parallel-guided mechanism is described, followed by finite element analysis (FEA) to validate the equations. We then describe the manufacture and testing procedure for a scaled parallel-guided mechanism which is used to experimentally validate the derived equations and FEA.

## Analytical derivation of stress scalability

**Displacement-driven compliant mechanisms.** A key condition that enables scalability is that the compliant mechanism be a displacement-driven mechanism. This means the device is designed to undergo a specified deflection (defined as a percentage of the flexure's length) without exceeding its yield strength [1]. This deflection also scales with the size of the device. In contrast to displacement-driven mechanisms, most mechanical components are designed to withstand a maximum force that may be applied during its function. For example, the structure of an aircraft wing is a traditional force-loaded system because it is designed to withstand the aerodynamic loads associated with flight. In such cases, the process involves designing the component to be both stiff and strong. Because stiffness and strength are commonly desired together, an incorrect intuition is often developed that equates stiffness and strength. However, the fact that stiffness and strength are different is essential to compliant mechanism design where components are intended to be strong yet flexible enough to withstand a desired displacement.

We demonstrate the theoretical difference between force-loading and displacement-loading of a compliant mechanism on the parallel-guiding mechanism in Fig 2A. This mechanism has a rigid shuttle and the direction of the applied force, $F$, as well as the symmetry of the system constrains the shuttle to translate without rotating. The two flexible (compliant) beams are fixed to the rigid shuttle at one end and to a rigid immovable ground at the other. The beams are made of the same material and have identical geometry.

In a traditional force-loaded mechanical component, a force is applied to the shuttle which induces stress on the beams. The equation for the yield stress of a fixed-guided beam is given in [1]:

$$S_y = \frac{F_{max}Lh}{4I} \tag{1}$$

Rearranging this equation for $F_{max}$ gives

$$F_{max} = \frac{S_ybh^2}{3L} \tag{2}$$

For a parallel-guided mechanism consisting of two fixed-guided beams, the maximum force (assuming small displacements) that this parallel-guided system can withstand before failing is therefore that which results in a maximum stress equal to the material strength, or

$$F_{max} = \frac{2S_ybh^2}{3L} \tag{3}$$

where $L$ is the flexible beams' length, $h$ is the beams' in-plane thickness, $b$ is their out-of-plane width, and $S_y$ is the material's strength (here defined as the maximum stress before failure). Larger forces can be held by increasing the stiffness, including by increasing the thickness ($h$) or width ($b$).

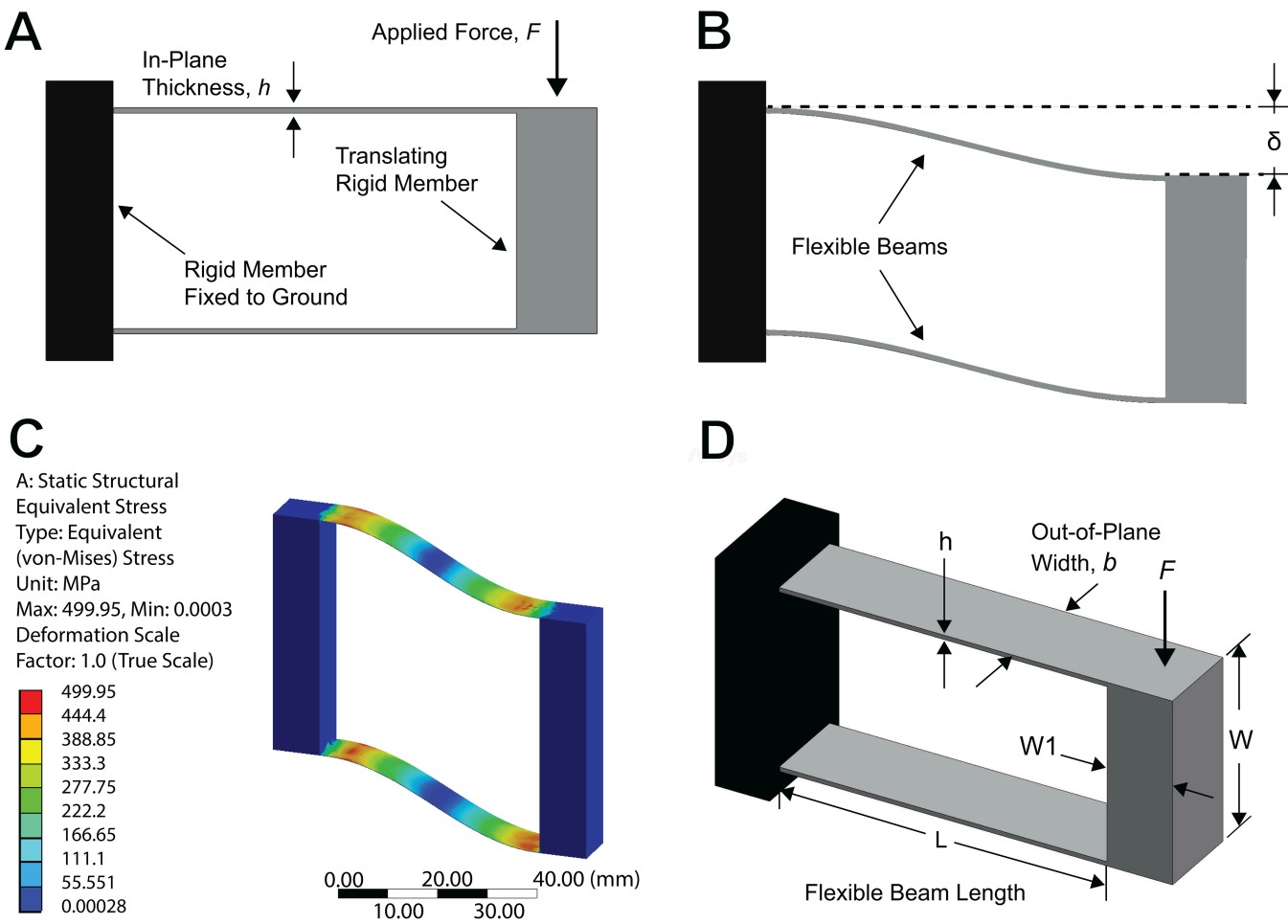

**Fig 2. A compliant parallel-guiding mechanism as an example of a scalable compliant mechanism.** (A) A compliant parallel-guiding mechanism with a specified applied force, $F$, (shown undeflected). (B) The same device shown as a displacement-driven compliant mechanism which has undergone a specified displacement, $\delta$. (C) The mechanism illustrated in an isometric view, with geometric parameters shown. (D) A prediction of stress for the parallel-guided mechanism (size A.1 from Fig 3) using finite element analysis software.

However, if the parallel-guiding mechanism is to be designed to undergo a maximum displacement without failure ($\delta_{max}$), then we begin with the equation for the maximum deflection for a flexible cantilever beam, retrieved from [1]:

$$\delta_{max} = \frac{2S_yL^2}{3Eh} \tag{4}$$

A parallel-guided mechanism consists of two fixed-guided beams. A fixed-guided beam has different boundary conditions than a cantilevered beam. However, a fixed-guided beam can be modeled by connecting the free ends of two half-length cantilever beams together. This means that the deflection for a fixed-guided beam can be described as

$$\delta_{max} = 2\frac{2S_y\left(\frac{L}{2}^2\right)}{3Eh} \tag{5}$$

And simplified as

$$\delta_{\max} = \frac{S_y L^2}{3Eh} \tag{6}$$

where $E$ is the material's Young's modulus. Note that the maximum displacement is increased by decreasing the thickness ($h$), which is the opposite of the change that would be made to withstand a larger force. But the differences are more than the thickness: of the five parameters in the equations ($L, h, b, S_y, E$), a change in any but the material strength results in different effects for force-loaded compared to displacement-driven systems. See Appendix 1 for more details on these derivations.

**Making stress independent of scale.** The ability to scale a displacement-driven compliant mechanism has a significant impact on the design of future compliant mechanisms. This unique trait is articulated and compared to other properties in Table 1.

We take the same parallel-guiding mechanism from Fig 2 and scale it by a scaling factor of $C$. For example, if $C = 2$, then the geometry and applied displacement are twice the magnitude of the original. The well-known cubed-squared law tells us that the mass will be $C^3$ larger than the original, while the surface area changes by a factor of $C^2$. The stiffness changes linearly and is $C$ times the original stiffness, while the natural frequency changes by the inverse of the scaling factor, or $1/C$ [18,31], and the strain energy stored in the deflected beams is scaled by $C^3$. Because these properties scale at different factors, it is challenging to scale a mechanical system from one size to another without significant changes to the geometry (see Table 1). However, there is one particularly important scaling effect for displacement-driven compliant mechanisms: the maximum displacement relative to the size, as measured by $\delta/L$, has a scaling factor of 1, meaning that it is the same regardless of the scale.

Scaling any displacement-driven mechanism by a factor of $C$ yields a scaled displacement $\delta^*_{max}/L$ of:

$$\frac{\delta^*_{\max}}{L} = \frac{S_y C L}{3ECh} \tag{7}$$

which simplifies to:

$$\frac{\delta^*_{\max}}{L} = \frac{\delta_{\max}}{L} \tag{8}$$

**Table 1. The result of scaling the geometry of the mechanism in Fig 2 by a factor of $C$ on different mechanical behaviors.** Note that the maximum deflection ratio before failure remains constant regardless of scaling the geometry.

| Property | Simplified Model | Resulting scaling factor if geometry scaled by C | Resulting scaling factor if b scaled by C |
|---|---|---|---|
| Mass ($M$) | $\rho[2hbL + w_1 w_2 b]$ | $C^3$ | $C$ |
| Surface area ($A$) | $2[2L(b+h) + w_1 w_2 + b(w_1 + w_2 - h)]$ | $C^2$ | N/A |
| Stiffness ($K$) | $\frac{24EI}{L^3}$ | $C$ | $C$ |
| Natural Frequency ($\omega$) | $\sqrt{\frac{K}{M}}$ | $\frac{1}{C}$ | 1 |
| Potential Energy ($U$) | $\frac{1}{2}K\delta^2$ | $C^3$ | $C$ |
| Maximum Force before Failure ($F_{max}$) | $\frac{2S_y bh^2}{3L}$ | $C^2$ | $C$ |
| Maximum Deflection Ratio Before Failure ($\frac{\delta_{max}}{L}$) | $\frac{S_y L}{3Eh}$ | 1 | 1 |

The consequence is that if the device size is halved, doubled, or scaled by some other desired factor, the $\delta_{max}/L$ remains the same.

The scaling factors presented in Table 1 were directly derived for the parallel-guiding mechanism, which has fixed-guided boundary conditions and experiences an applied deflection at the end of the beams. While the Euler-Bernoulli beam theory equations are slightly altered based on the boundary and loading conditions, when dividing the model representing the scaled geometry by the model representing the unscaled geometry, the changes cancel, leaving the same resultant scale factors. These scaling factors are therefore applicable to displacement-driven compliant mechanisms in general.

Displacement-driven compliant mechanisms may be actuated through a variety of methods [12], including cable-driven, shape-memory alloy, piezoelectric, magnetic, fluidic, pneumatic, or resorbable materials [32]. For the mechanisms developed here, we generally used more manual methods, e.g. actuation through manual application of deflection. Regardless of the actuation method selected, since smaller-scale mechanisms experience decreased stiffness, potential energy, and total deflection (as shown in Table 1), they also require less power to experience their full actuation. The degrees of freedom of a compliant mechanism would also remain constant as it is scaled [33–35].

**Making stress independent of out-of-plane thickness.** The other remarkable consequence of implementing a displacement-driven system is that the maximum allowable displacement before material failure from stress is independent of the width, $b$. For example, if the width of the flexible beam in the parallel-guiding mechanism is doubled or quadrupled (Fig 3C), the maximum allowable displacement does not change. However, the amount of force required to achieve that displacement is linearly proportional to the width. This has significant implications for device design. The geometry of displacement-driven compliant mechanisms can be designed using the thickness and length to achieve the desired displacement and then the width can be selected to achieve the desired actuation force without affecting the stress. This is nonintuitive and counter to what occurs with a force-loaded system. This means that a compliant mechanism such as a one-piece compliant latch (imagine simple consumer devices such as a backpack latch or snap connectors for electronics assembly) can be designed by selecting the best length and thickness to obtain a manufacturable device that fits in the geometric and material constraints and then the width can be selected to achieve the desired reaction force.

**Stress scalability under torsion.** In addition to elements that bend, compliant mechanisms may use elements in torsion to achieve motion with a prescribed angular displacement. The maximum angle of twist ($\theta$) for a square cross section beam with side lengths $b = h$ is described as

$$\theta = \frac{1.475\tau_{max}L}{hG} \tag{9}$$

where $\tau_{max}$ is the maximum allowable shearing stress, $L$ is the length of the torsional element, and $G$ is the shear modulus. To determine the maximum angle of twist for other rectangular cross sections, the constant 1.475 in Eq 9 would be replaced by the quotient of $\alpha$ and $\beta$ for the corresponding $b/h$ rectangular profile (where $b > h$) in Table 2. Displacement-driven compliant mechanisms with elements in torsion exhibit the same scaling trait as those with elements in bending: if the geometry is scaled by a factor of $C$, the resulting scaling factor is 1. This means the maximum angle of twist remains the same for a proportional increase in $L$ and $h$. Furthermore, this angular displacement is achieved with the same maximum shearing stress value regardless of scale.

**Table 2. Values for $\alpha$ and $\beta$ as functions of $b/h$.** This table is adapted from [36].

| $b/h$ | 1.00 | 1.50 | 1.75 | 2.00 | 2.50 | 3.00 | 4.00 | 6.00 | 8.00 | 10.00 | $\infty$ |
|---|---|---|---|---|---|---|---|---|---|---|---|
| $\alpha$ | 0.208 | 0.231 | 0.239 | 0.246 | 0.258 | 0.267 | 0.282 | 0.299 | 0.307 | 0.313 | 0.333 |
| $\beta$ | 0.141 | 0.196 | 0.214 | 0.228 | 0.249 | 0.263 | 0.281 | 0.299 | 0.307 | 0.313 | 0.333 |

## Manufacture and testing of scaled parallel-guiding mechanisms

To verify the effect described by this analytical model, a series of various scales of parallel-guided mechanisms were designed, evaluated in finite element analysis software, manufactured (shown in Fig 3), and experimentally tested. The mechanisms were designed with common sheet material thicknesses and off-the-shelf parts to facilitate production in multiple scales. Their dimensions are shown in Fig 3D. The flexible segments of the parallel-guided mechanisms were made from 1095 blue-tempered spring steel. The rigid members were made from aluminum for the fully-scaled mechanism or 3D printed PLA plastic for the width-scaled mechanism. The fasteners are made from zinc-plated steel.

**Finite element analysis validation of scaled parallel-guiding mechanism.** Finite element analyses of simplified parallel-guided mechanisms were performed (see Fig 4) using commercially available software (ANSYS Workbench). The primary purpose of these FEA is to assess the stress distribution and deformation characteristics of the flexible

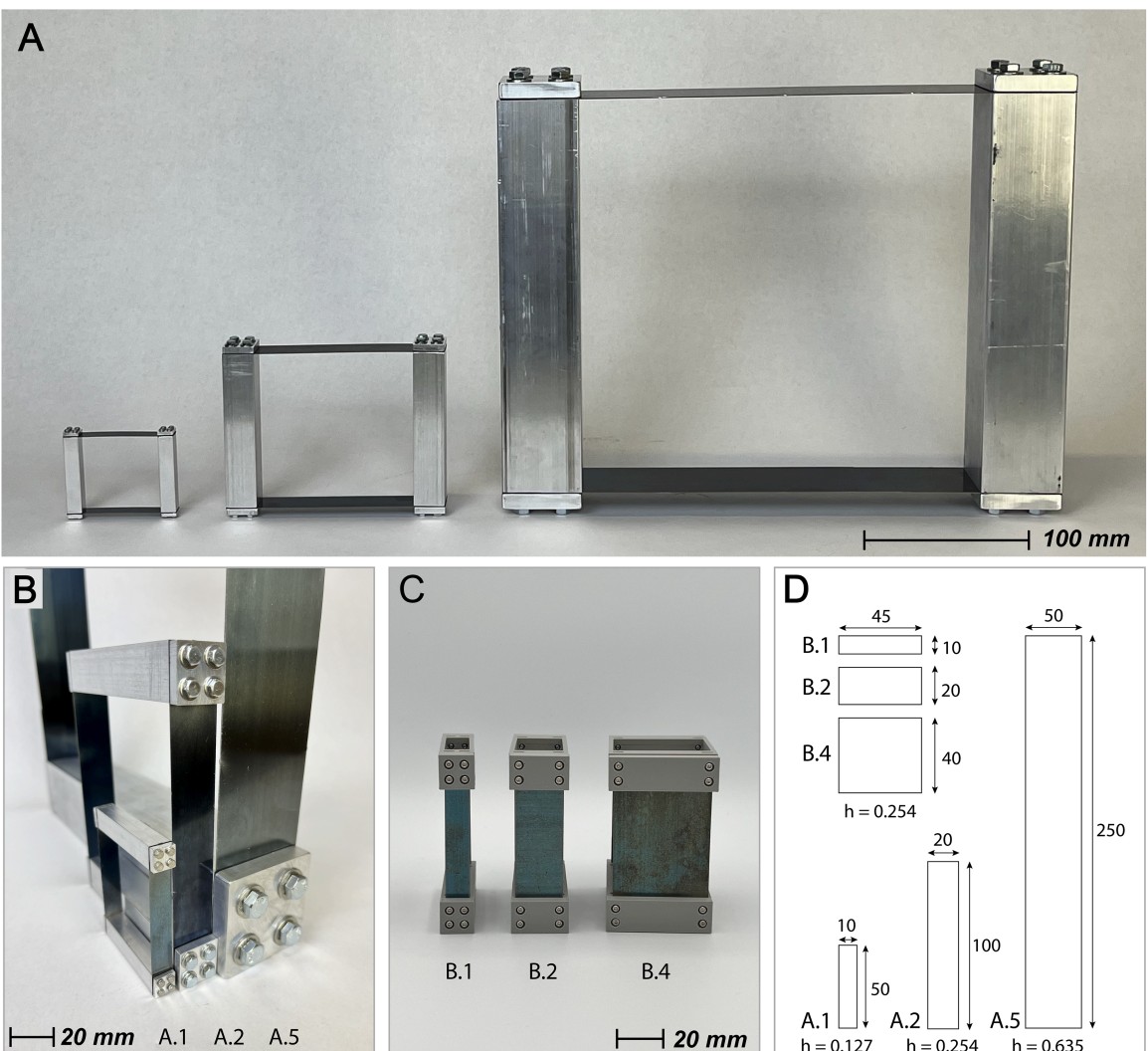

**Fig 3. Compliant parallel-guiding mechanisms are shown at various scales.** (A-B) Three parallel guiding mechanisms with increasing values of the scaling factor, C, (from left to right: C = 1, C = 2, C = 5). (C) Three parallel guiding mechanisms with increasing values of the scaling factor for the width b (from left to right: C = 1, C = 2, C = 4). (D) The dimensions of A.1, A.2, A.5, B.1, B.2, and B.4 are illustrated (not to scale). All dimensions are in units of millimeters. For both Series A and B, the number in the label represents the relative scaling factor from the first mechanism in the series.

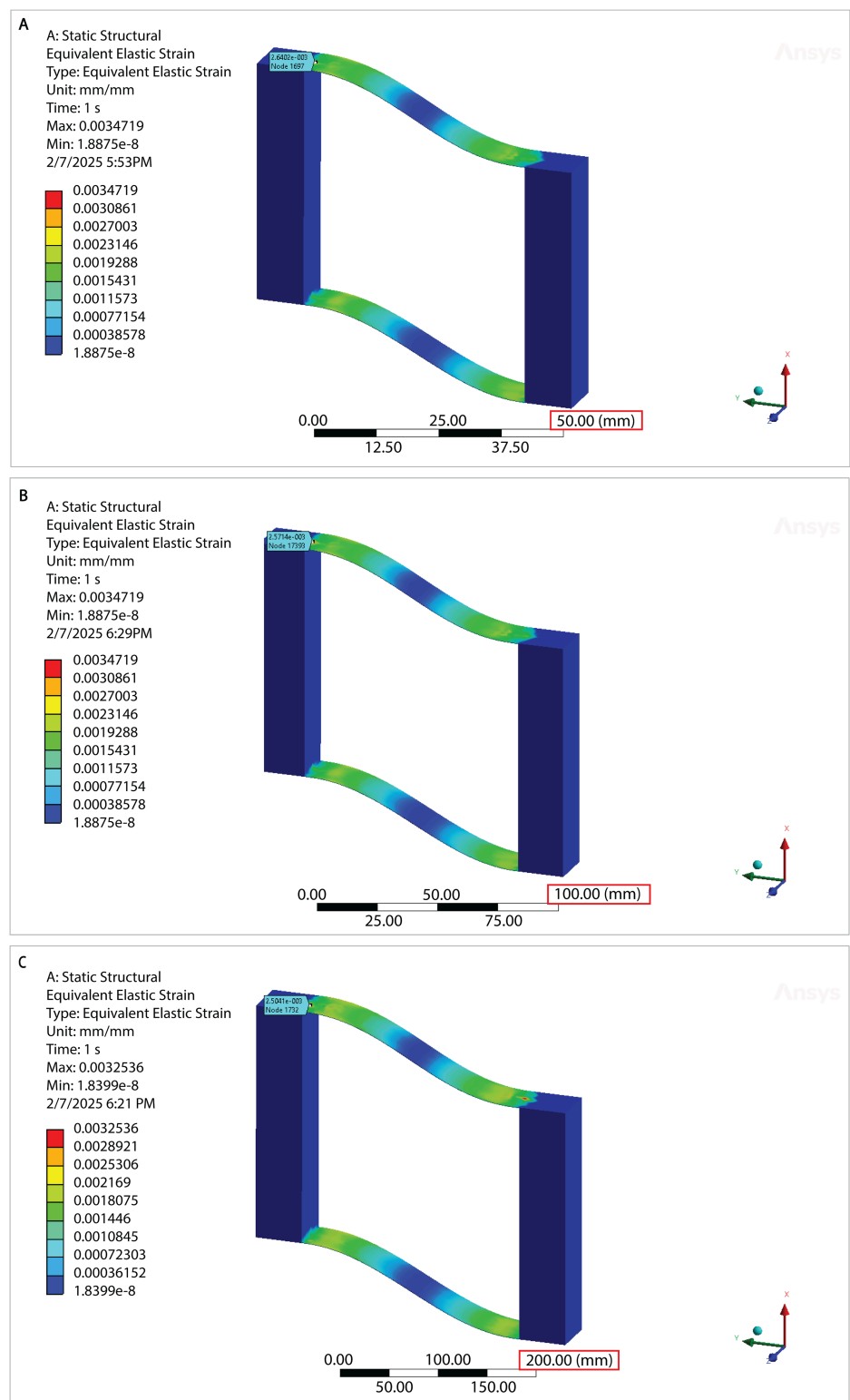

**Fig 4**. **Finite element analyses of strain in three scales of a compliant parallel-guided mechanism.** Finite element simulations of each parallel-guided mechanism scale shown in Fig 3A and 3B. The observed strain values aligned with the experimental data shown in Fig 6.

members under an applied displacement load. The simulated geometry has the same critical dimensions as the test specimens A.1, A.2, and A.5 shown in Fig 3A and 3B. All geometric features pertinent to the stress evaluation are present in the model; fasteners and clamping plates were omitted from the simulation model. The material properties for 1095 Spring Steel were used: Young's Modulus = 205 GPa, Poisson's Ratio = 0.29, Bulk Modulus = 1.627e11 Pa, Shear Modulus 7.946e11 Pa, Tensile Yield Strength = 800 MPa. The ANSYS Mechanical meshing tool was used to generate a mesh with the SOLID187 element type and refinement on the area of interest (the flexible members). Boundary conditions and constraints include a fixed support on one of the rigid members, and a 0.3 ($\delta/L$) displacement load on the other rigid member, in the direction parallel to its length. The solver type is the Static Structural Solver. The outputs included "Total Deformation" and "Equivalent Elastic Strain".

The finite element models display a nearly identical pattern for each size. This observation advances the notion that strain and stress remain independent of scaling factor for deflected beams; experiments on flexure notch hinges demonstrate a similar conclusion [10].

**Experimental validation of scaled parallel-guiding mechanism.** Further verification of the scaling trait was obtained by placing the physical hardware shown in Fig 3 in a tensile test machine to investigate the mechanisms' force/displacement behavior. A schematic of the testing setup is illustrated in Fig 5. The compliant parallel-guided mechanism test specimen was placed such that one rigid component was fixed (relative to the stationary body of the tensile tester) and the other was free to slide over rollers as actuated by a probe attached to the load cell. The tensile tester machine used for these tests is a 4.448 kN "Instron" tensile tester from *Illinois Tool Works Inc.* A 24.91 N force transducer from *Interface Inc.* was used to obtain force readings.

A physical calibration method was used to ensure accurate load cell measurements. The load cell was first connected to the Data Acquisition system (DAQ) and zeroed. A dead weight of known magnitude was placed on the load cell to confirm the measurement was as expected. The tensile tester was rotated such that the direction of travel was perpendicular to gravity. A lightweight probe was attached to the load cell and applied a small moment to it, but the structural stiffness of the load cell rendered the effect too slight to consider. The load cell was kept at the same calibration state throughout trials; there were no software calibrations in between tests on the parallel-guided mechanism specimens.

The three sizes of compliant parallel-guided mechanisms were machined, assembled, and equipped with BFH350-3AA high-precision foil strain gauges (see Fig 5C). The strain gauges have physical dimensions as follows: Basal Size

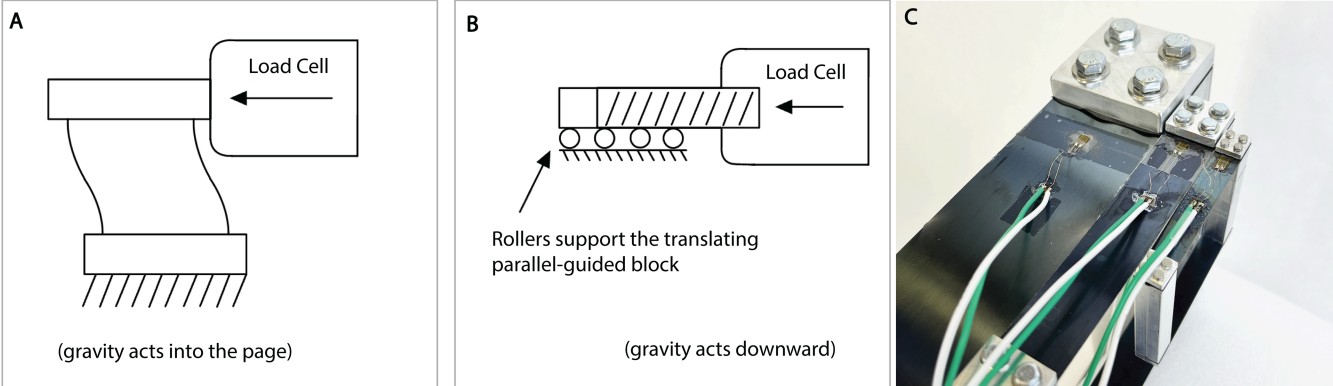

**Fig 5**. **The testing setup for compliant parallel-guided mechanism data collection.** (A-B) Two views of the testing setup used for both the force/displacement and strain tests are illustrated here. (C) High-precision strain gauges on flexible beams of three geometric scales of a compliant parallel-guided mechanism. Strain gauges were adhered to the Series-A parallel guided mechanisms. The smallest specimen, A.1, has a strain gauge touching the end of the flexible member—the point of maximum bending stress. To make an approximate comparison between scales, the center of the wire grid for each strain gauge is located at the same relative distance along the length of the flexure for each specimen.

(7.2 mm x 4.1 mm), Wire Grid (3.2 mm x 3.1 mm). The strain gauges' nominal resistance is 350 ± 1 Ω. The strain gauges were adhered to the blue tempered spring steel via a cyanoacrylate compound.

This scaling effect is demonstrated in more complex systems later on. Some of these systems are demonstrated at such significantly different scales that it becomes impossible to manufacture them from the same material. It is therefore important to note that the scaling effect requires that the ratio of the strength to modulus ($S_y/E$) is approximately the same (or higher) across different scales. This is straightforward for scaling factors where the same material can be used, but more care is needed to maintain (or improve) this ratio when scaling requires different materials or different fabrication processes.

## Results

### Parallel-guiding mechanism

The results from tensile tests are shown in Fig 6. These data show the strain response for the three fully-scaled parallel-guided mechanisms (top) and the force/displacement response for the three width-scaled parallel-guided mechanisms (bottom). For the fully scaled mechanisms, the strain is visually essentially the same between scales. There is slightly more variation seen in the force values measured for the width scaled mechanisms, but they still show similar results across scales. These results validate the theory presented above: that strain remains constant as geometry is scaled, and that the mechanism's force is linearly proportional to its width.

### Case study: Scaling beams in bending

The principles of scalability can be demonstrated in more complex configurations than a parallel-guided mechanism. Consider the commercially available toy projectile launcher shown in Fig 7A (disassembled) and Fig 7B (assembled), which has over eighty separate parts. Its primary function was replaced with the resulting monolithic (single piece) compliant mechanism shown in Fig 7C. (It is worth noting that replacing eighty parts with a single-piece compliant mechanism is in itself a feat of engineering design but is not the subject of this paper.) The device stores strain energy in the flexible beams of a compliant mechanism and then transfers that energy to kinetic energy of a projectile. The phenomena described previously can be exploited to create a system with the same geometry at multiple scales. The materials may be different at these different scales, so the strength and Young's modulus also change, but are still in a range that enable the effect to be demonstrated.

The flexible beams are oriented such that they store mechanical strain energy in bending and in compression. The maximum displacement is controlled by hard stops in the system. A latching feature is connected to a flexure that acts as a trigger mechanism, releasing the strain energy in the slanted beams when activated by an applied force. Once released, the shuttle returns to its low energy state and impacts the projectile, propelling it forward (the most kinetic energy is transferred from shuttle to projectile when the mass of the projectile approaches the mass of the moving shuttle system, a principle also illustrated by a "Newton's Cradle").

Once the system has been designed to undergo the desired displacement, with enough margin to account for changes in materials at other scales, the principles described here enable the system to be scaled to different sizes without mechanism failure. An important result of this is that the device can be manufactured at sizes that correspond to the available manufacturing process. Consider the differences in work area for an industrial laser cutter, a tabletop 3D printer, and a photolithography system. At each macro, meso, and micro scale, the displacement-driven projectile-launcher design can be produced through a straightforward scaling of its geometry to the appropriate work area size.

This compliant mechanism was fabricated and tested at the $C = 1$ scale—the scale of a commercially available toy dart launcher—from plastic using additive manufacturing (Fig 7C) and aluminum via waterjet cutting (Fig 7D). The 1/10th and 1/100th scales (Fig 7D) were manufactured from a carbon-infiltrated carbon nanotube forest [20,37–39]. The 7.25 times scale was milled from a rectangular polypropylene sheet (Fig 7D). Functionality was demonstrated at each of these levels

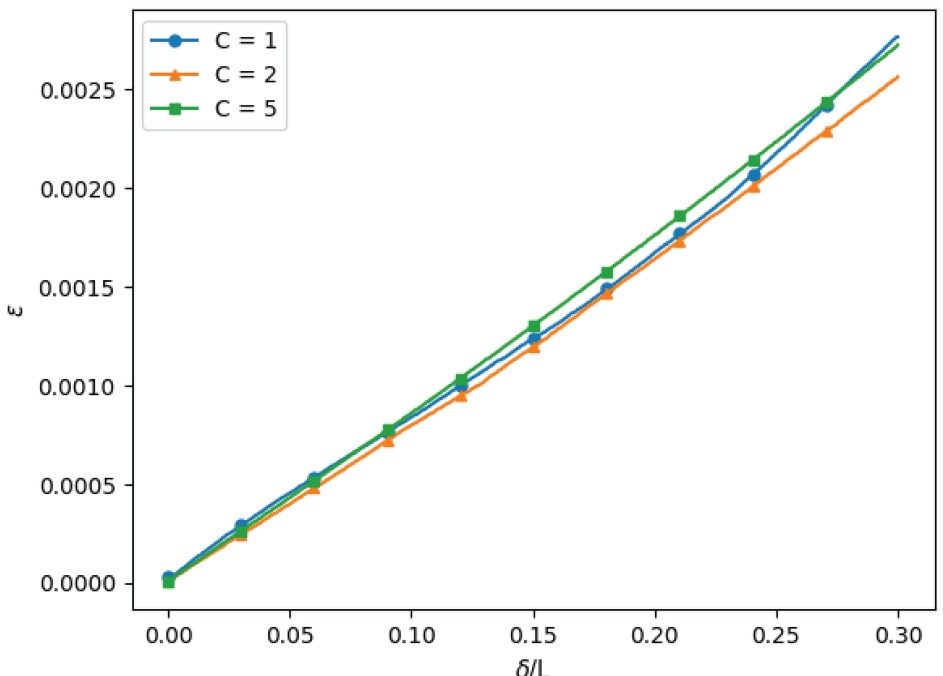

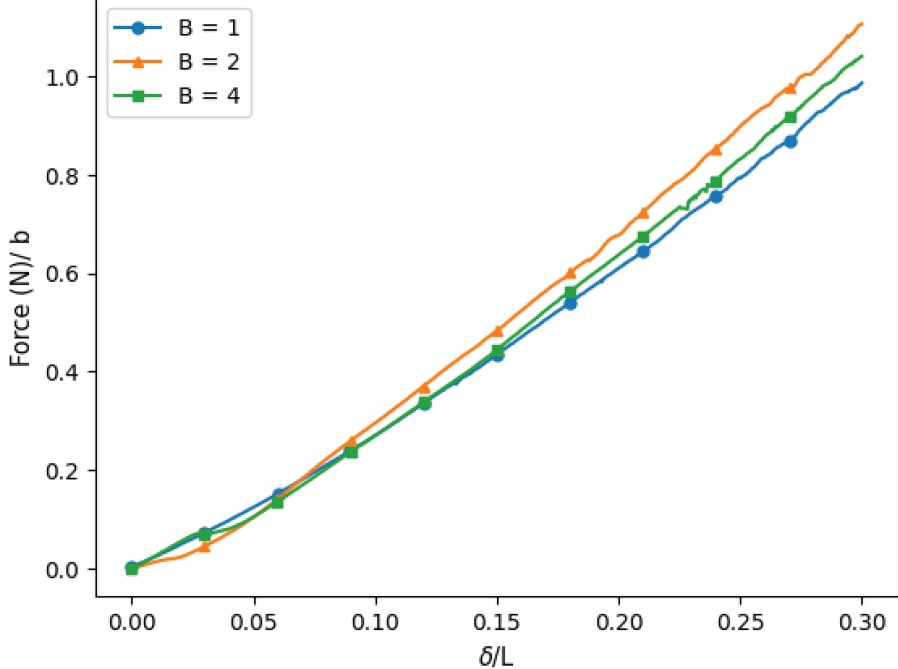

**Fig 6**. **Strain and force data is collected for various scales of compliant parallel-guiding mechanisms.** Top: The measured strain ($\epsilon$) for the displacement-driven compliant mechanism at different scales. Bottom: The reaction force (F) over the width $b$ for three different scales of $b$ from a specified input displacement range of 0.0 to 0.30 $\delta/L$.

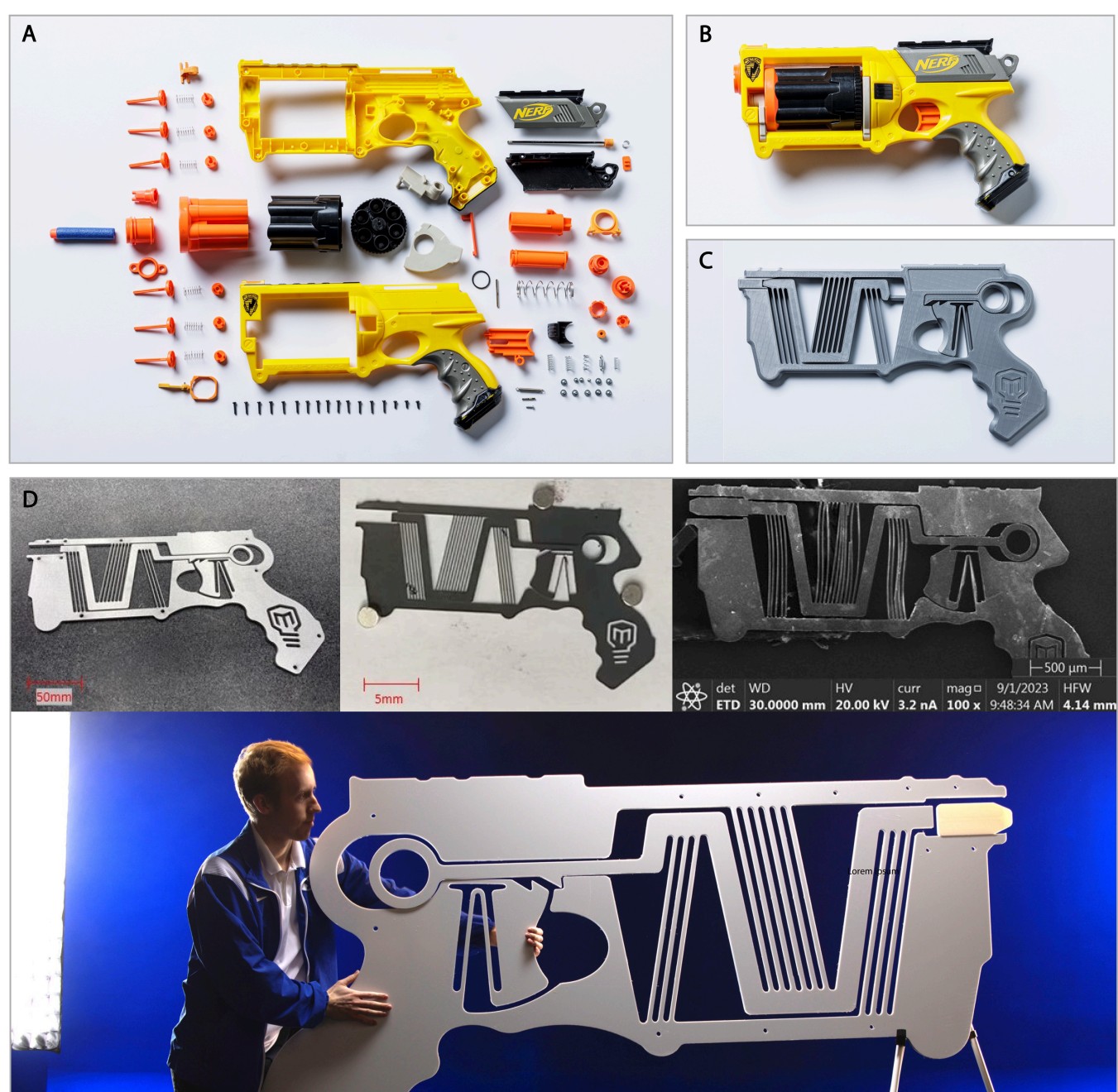

**Fig 7**. **A toy dart launcher as a demonstrative compliant mechanism.** (A) A disassembled commercially available foam dart launcher illustrating that over eighty parts are used to construct the device. (B) An assembled foam dart launcher. (C) A monolithic compliant mechanism version of the launcher made from a single 3-D printed piece. (D) The compliant launcher at 100% of the scale of the original launcher ($C = 1$), at one-tenth scale ($C = 0.1$), one-hundredth scale ($C = 0.01$), and 725% of the original ($C = 7.25$) with a person for reference. The individual in this figure has given written informed consent (as outlined in PLOS consent form) to have their image appear in this paper.

and in the associated materials (spring steel, polypropylene, carbon-infiltrated nanotubes, PLA plastic). The toy blaster was also manufactured from high-carbon steel, Baltic birch plywood, amorphous metallic glass, fiberglass, aluminum, acrylic, and low-carbon steel.

Unique to this device, however, is that it was informally validated by tens of thousands of downloads, fabrication, and tests of the device at multiple scales throughout the world (see Appendix S1 for additional information).

Carbon Infiltrated Carbon Nanotubes (CICNTs) were used to create several versions of the projectile launcher shown in Fig 7D. CICNTs are stronger than standard carbon nanotubes which is important for compliant mechanisms that have to perform well under bending [39,40] (see Appendix S2 for the generalized procedure used to create these CICNT mechanisms). A Keyence VHX microscope was used in conjunction with a micromanipulator probe to view and actuate the CICNT devices, as shown in video for the 1/100th scale in [30].

When manufacturing at extreme scales, such as the $C = 0.01$ and $C = 7.25$ scales of this projectile launcher, some difficulties arise. For example, at the microscale, the smallest feature of the projectile launcher is less than 100 $\mu$m. Manufacturing long slender beams for bending with this width while avoiding failure, even with CICNTs, can be difficult and requires extreme precision and carefully selected parameters in manufacturing [20,41,42]. This is also linked to material limitations: research in the area of improving mechanical properties of CICNTs often focuses on increasing strength and modulus [43,44], while compliant mechanisms applications benefit instead from a lower modulus [1]. Additionally, manufacture at the micro scale can be more time and resource intensive than at the macro scale. On the mega scale, in extreme cases, it can become difficult to obtain sheet material or manufacturing facilities large enough to create the mechanism in a single piece, without requiring assembly. This can also become more time and cost intensive.

The $C = 1$ scale additively manufactured launcher shown in Fig 7C is able to launch the dart approximately 4 m from a standing position. The CICNT 1/10th scale as shown in Fig 7 is able to launch the dart approximately 10 cm from a position flat on the table. The CICNT 1/100th scale is able to launch the dart approximately 3 cm from flat on the table. An example of each of these sizes launching, including visuals of the torsional beams in deformation, is available in [30].

## Case study: Scaling beams in torsion

The Lamina Emergent Torsion (LET) joint is used to demonstrate the scaling conditions for systems loaded in torsion. The LET joint relies on torsional members to achieve a prescribed displacement [45] and an array of LET joints and an individual LET joint are illustrated in Fig 8A. In Fig 8B, a flat sheet contains hundreds of LET joints in specific locations that enable it to fold from a two-dimensional sheet into the three-dimensional form of a chair [46]. See Appendix S3 for a video showing the manufacture and actuation of this device. This process has been utilized to develop other lamina emergent mechanisms and structures as well [1,47,48]. These LET surfaces are arrays of LET joints in series and parallel. The geometry of the LET joints is designed such that no single torsional member of the array exceeds its maximum angle of twist (see Fig 8A). The LET surfaces are functional (they resist out of plane forces) and aesthetically intriguing (Fig 8C).

Once the chair has been designed with the appropriate overall angles (backrest, seat, chair legs, etc.) and LET joints of corresponding size (length and width), the chair can be scaled to various sizes without changing the size or placement of the LET joints. Note that as the overall length and width of the flat sheet stock material is scaled, its thickness must be scaled proportionally as well. Three scales of a compliant chair were created at one-third, one-half, and full scale (see Fig 8D). The chairs in Fig 8D are fabricated from fluorescent acrylic sheet, but they have also been demonstrated in Baltic birch plywood [46]. Parkinson et al. performed modeling on the birch plywood chair which demonstrated that it could withstand 4,000 N before failure. The acrylic version is expected to be able to withstand a lower force but is verified for at least 1,000 N through usability testing.

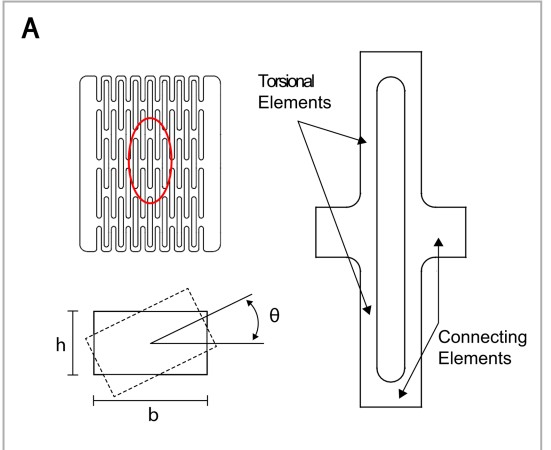
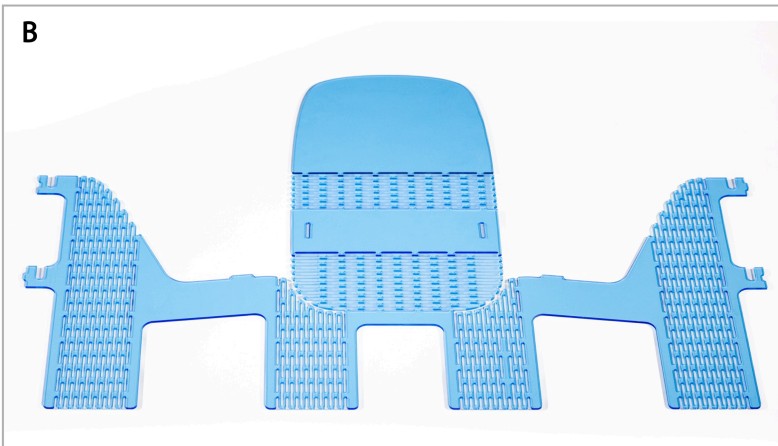
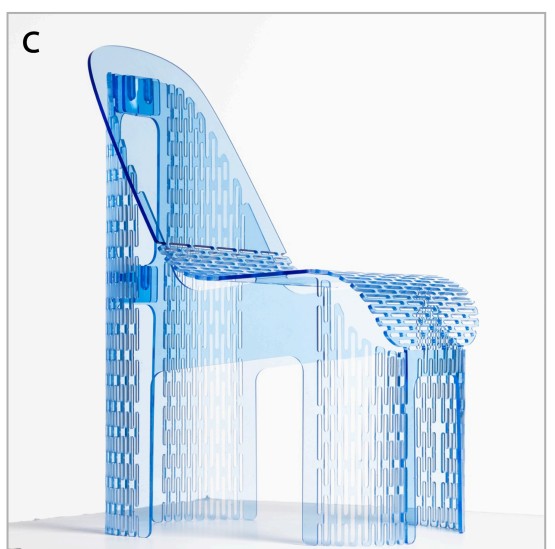
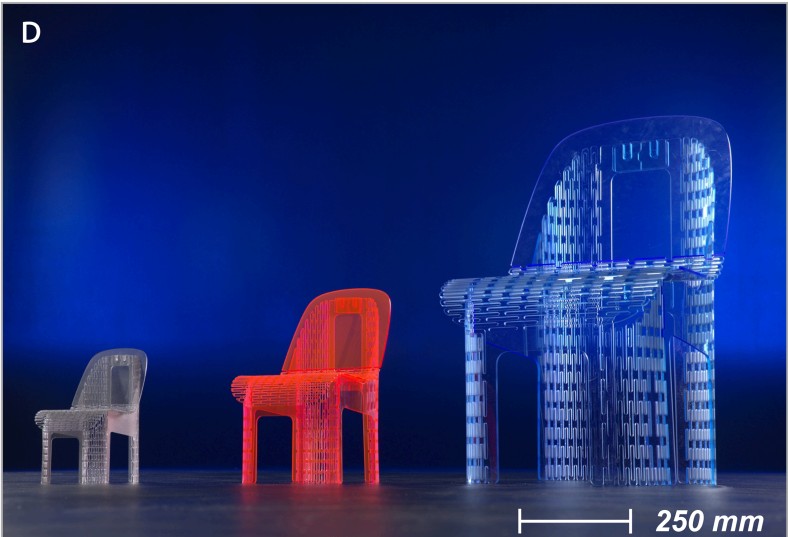

**Fig 8**. **A monolithic chair folded from a single piece of material.** (A) A lamina emergent torsional (LET) joint as a unit cell used in the array. (B) A single piece of fluorescent acrylic with cuts made to create arrays of LET joints in select locations. (C) The 1.22 m by 1.83 m acrylic sheet folded into a chair. (D) Three sizes of the chair including a doll-house chair ($C = 1/3$), a child-size chair ($C = 1/2$), and an adult-size chair ($C = 1$).

## Discussion

The behavior of displacement-driven compliant mechanisms is counterintuitive to those accustomed to designing force-loaded mechanical systems. This makes compliant mechanisms more challenging to design but also has the potential for creating new systems with unprecedented performance. For example, consider the possibility of medical implants or prosthetic limbs that replace damaged or diseased human anatomy with compliant mechanisms with similar size and functionality as the natural system [16,49–51] and can be scaled to accommodate different body sizes. This can be particularly valuable for medical implants or prostheses for adolescents who experience significant size changes due to growth. This principle could also be extended to the design of surgical tools that utilize compliant mechanisms to position, sense, grasp, cut, or transmit force [12,52–54]. A consistent design across tool sizes would likely aid a surgeon tasked with performing work on large and small bodies.

Similarly, in robotics applications, compliant mechanism bearings [55,56] can be designed for the desired rotational motion and easily scaled to accommodate other sizes, or the width modified to provide the desired return-to-home spring

constant. Scientific instruments can be designed that will retain their function at various scales while maintaining consistent relative geometry [57,58]. Additionally, metamaterial concepts are often developed and demonstrated at one size but ideally are scalable to others [59–62]. Implementing the concepts presented here could facilitate this scaling.

Existing compliant mechanisms design frameworks typically include a definition of the desired behavior, concept selection, virtual prototyping, followed by physical prototyping [1,63]. Without knowledge of the scaling principles presented here, this design process would essentially need to be performed once for each scale of the mechanism. By implementing scaling principles into this existing framework during the first step of the design process, designers can instead perform the design process a single time for multiple scales and would need to consider tuning for the manufacturing process and material properties of the resultant design to achieve performance at differing scales. This has the added benefit of simplifying prototyping for extreme scales by allowing it to be done primarily at a macro scale, where lessons learned from can then be applied when manufacturing the final products in the micro or mega scales.

When scaling displacement-driven compliant mechanisms, it is also important to consider challenges for manufacturing at micro or mega scales. Manufacturing at the micro-scale is possible through limited methods and is typically slower, more expensive, and requires more complex systems than at the macro-scale [21], particularly compared to macro-scale rapid prototyping methods. For complex systems that are expected to achieve motion through compliant members, such as the toy projectile launcher discussed in the case study above, the micro-scale manufacturing options become even more limited [11]. Fabrication tolerances also scale approximately with the mechanism, so what might start as a tolerance of 1 mm at the $C = 1$ scale might become a tolerance of a 10 $\mu$m at the $C = 0.01$ scale, which again increases manufacturing difficulty.

It is also important to consider non-ideal material properties and the potential for stress concentrations when working with extreme scales. A tiny material nonideality or stress concentration that might in a certain material or scale not affect performance could in another material or scale decrease performance or even lead to catastrophic failure. Additionally, for mechanisms undergoing repeated cyclic motion, which compliant mechanisms often require, a consideration of fatigue behavior is also important. As with stress concentrations, we see similarly magnified effects in fatigue life: what could have a minimal effect in one material or scale, might significantly reduce fatigue life in another material or scale.

By designing systems that are readily scalable, custom-sized products, which would otherwise be economically inefficient, become financially viable given that the design cost occurs only once [64]. Displacement-driven compliant mechanisms offer a significant contribution to the world's growing capability to provide highly personalized solutions to individual needs on a wide-reaching and cost-effective basis.

## Conclusion

The non-scaling property of stress as the geometry is scaled provides unique opportunities for developing mechanical systems that can be used at multiple scales. It also means that prototypes can be designed, fabricated, and tested at the most convenient or cost-effective size, then scaled to and refined at the final size—similar to nondimensional numbers used for testing flight vehicles [65]. While other design adjustments may be needed due to manufacturing and material constraints at different levels, the stress being independent from scale is a significant, simplifying factor in the overall system development.

Scaling a physical device's geometry results in mechanical properties changing in various ways (e.g. the cubed-squared law states that for a scaling factor $C$, mass scales with $C^3$ and surface area with $C^2$). These scaling effects can result in a device's inconsistent and unplanned mechanical behavior when varying its fabricated size, thereby necessitating unique designs at different scales. We show that for displacement-driven compliant mechanisms, mechanical stress is uniquely invariant with scale. This effect is described theoretically, verified through computer models and physical testing, and is demonstrated in three examples: a parallel-guiding mechanism, a projectile launcher, and a deployable chair.

This enhanced understanding of stress invariance provides innovative insight into the way devices can be designed for systems that operate across different scales.

## Supporting information

**S1 Appendix. Supplementary text.** Information regarding the force/displacement and strain data collection [66], the supplemental movie documentaries, and the derivation of equations is described in this appendix.
(DOCX)

**S2 Appendix. Materials and methods.** A brief description of the carbon infiltrated carbon nanotube manufacturing process is given.
(DOCX)

**S3 Appendix. Video.** A video demonstrating the chair manufacture and assembly is provided. The individuals in this video have given written informed consent (as outlined in PLOS consent form) to have their images appear in this paper.
(MOV)

## Acknowledgments

The contributions of Terri Bateman, Kevin Cole, David Morgan, Denise Halverson, Carolina Wright, Grant Ogilvie, Bridgette Kemper, Aliya Bascom, James Wade, Hunter Pruett, Natalie Jaques, Luke Gardner, Ivyann Running, Michael Linder, Brooklyn Peters, Jared Erickson, Austin Martel, Lydia Beazer, Trevor Carter, Audrey Christiansen, Davis Wing, Brooklyn Clark, Andrew Geyser, Corinne Jackson, Samuel McKinnon, Lais Oliveira, Felipe Rivera, Spencer Shirley, Jake Sutton, Nathan Coleman, Philip Klocke, Clark Roubicek, Mitchel Skinner, Katie Varela, and Kyle Dahl, for their work in analysis, prototype fabrication, and testing, are greatly appreciated. We acknowledge Robert Davis and Richard Vanfleet for their work in developing carbon nanotube fabrication technology that was used in this project. Other contributions to manufacturing include Boston Micro Fabrication, with micro-3D printing; FineArc Welding LLC with waterjet cutting; and the Center for Medical Innovation at the University of Utah with Fused Deposition Modeling of nylon parts. We thank BYU Photo for photography services.

## Author contributions

**Conceptualization:** Jared R. Hunter, Mark B. Rober, Brian D. Jensen, Spencer P. Magleby, Larry L. Howell.

**Funding acquisition:** Mark B. Rober.

**Investigation:** Jared R. Hunter, Bethany Parkinson, Jacob L. Sheffield.

**Methodology:** Jacob L. Sheffield, Mark B. Rober, Brian D. Jensen, Nathan S. Usevitch, Larry L. Howell.

**Project administration:** Mark B. Rober, Brian D. Jensen, Nathan S. Usevitch, Larry L. Howell.

**Validation:** Jared R. Hunter, Jacob L. Sheffield.

**Visualization:** Jared R. Hunter, Mark B. Rober, Brian D. Jensen.

**Writing – original draft:** Jared R. Hunter, Bethany Parkinson, Jacob L. Sheffield, Brian D. Jensen, Larry L. Howell.

**Writing – review & editing:** Jared R. Hunter, Bethany Parkinson, Brian D. Jensen, Spencer P. Magleby, Larry L. Howell.

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
