## [Decision Letter · Decision Letter 0]

11 Jul 2025

PONE-D-25-26746 Monolithic scalable compliant mechanisms PLOS ONE

Dear Dr. Parkinson,

Thank you for submitting your manuscript to PLOS ONE. After careful consideration, we feel that it has merit but does not fully meet PLOS ONE’s publication criteria as it currently stands. Therefore, we invite you to submit a revised version of the manuscript that addresses the points raised during the review process.

Based on the reviewers' comments and our evaluation of the paper, we recommend a major revision.

We kindly ask the authors to make detailed revisions in line with the reviewers' suggestions.

We look forward to receiving your revised manuscript.

Kind regards,

Geng Wang, PhD

Academic Editor

PLOS ONE

Journal Requirements:

[This work was supported through funds awarded to BDJ by Castle Grayskull, LLC, owned by Mark Rober. Mark Rober did not participate in data collection and analysis. He did participate in study design, the decision to publish, and preparation of the manuscript.].

We note that one or more of the authors is affiliated with the funding organization, indicating the funder may have had some role in the design, data collection, analysis or preparation of your manuscript for publication; in other words, the funder played an indirect role through the participation of the co-authors. If the funding organization did not play a role in the study design, data collection and analysis, decision to publish, or preparation of the manuscript and only provided financial support in the form of authors' salaries and/or research materials, please do the following:

1. Review your statements relating to the author contributions, and ensure you have specifically and accurately indicated the role(s) that these authors had in your study. These amendments should be made in the online form.

2. Confirm in your cover letter that you agree with the following statement, and we will change the online submission form on your behalf:

“The funder provided support in the form of salaries for authors [insert relevant initials], but did not have any additional role in the study design, data collection and analysis, decision to publish, or preparation of the manuscript. The specific roles of these authors are articulated in the ‘author contributions’ section.

4.  We note that you have a patent relating to material pertinent to this article. Please provide an amended statement of Competing Interests to declare this patent (with details including name and number), along with any other relevant declarations relating to employment, consultancy, patents, products in development or modified products etc. Please confirm that this does not alter your adherence to all PLOS ONE policies on sharing data and materials, as detailed online in our guide for authors http://journals.plos.org/plosone/s/competing-interests by including the following statement: ""This does not alter our adherence to  PLOS ONE policies on sharing data and materials.” If there are restrictions on sharing of data and/or materials, please state these. Please note that we cannot proceed with consideration of your article until this information has been declared.

[The contributions of Terri Bateman, Kevin Cole, David Morgan, Denise Halverson, Carolina Wright, Grant Ogilvie, Bridgette Kemper, Aliya Bascom, James Wade, Hunter Pruett, Natalie Jaques, Luke Gardner, Ivyann Running, Michael Linder, Brooklyn Peters, Jared Erickson, Austin Martel, Lydia Beazer, Trevor Carter, Audrey Christiansen, Davis Wing, Brooklyn Clark, Andrew Geyser, Corinne Jackson, Samuel McKinnon, Lais Oliveira, Felipe Rivera, Spencer Shirley, Jake Sutton, Nathan Coleman, Philip Klocke, Clark Roubicek, Mitchel Skinner, Katie Varela, and Kyle Dahl, for their work in analysis, prototype fabrication, and testing, are greatly appreciated. We acknowledge Robert Davis and Richard Vanfleet for their work in developing carbon nanotube fabrication technology that was used in this project. Other contributions to manufacturing include Boston Micro Fabrication, with micro-3D printing; FineArc Welding LLC with waterjet cutting; and the Center for Medical Innovation at the University of Utah with Fused Deposition Modeling of nylon parts. We thank BYU Photo for photography services. Funds for this work were provided by Castle Grayskull, LLC.]

[This work was supported through funds awarded to BDJ by Castle Grayskull, LLC, owned by Mark Rober. Mark Rober did not participate in data collection and analysis. He did participate in study design, the decision to publish, and preparation of the manuscript.]

6. We note that Figure 1 and 9 includes an image of participant in the study.

7. Thank you for stating the following in the Competing Interests section:

[I have read the journal's policy and the authors of this manuscript have the following competing interests: This work was supported through funds awarded to BYU by Castle Grayskull, LLC, owned by Mark Rober. A patent application is pending on the planar spring that is integral to the blaster case study. A patent has been issued on the LET joints that are part of the chair case study.].

We note that one or more of the authors are employed by a commercial company: name of commercial company.

Within your Competing Interests Statement, please confirm that this commercial affiliation does not alter your adherence to all PLOS ONE policies on sharing data and materials by including the following statement: ""This does not alter our adherence to  PLOS ONE policies on sharing data and materials.” (as detailed online in our guide for authors http://journals.plos.org/plosone/s/competing-interests). If this adherence statement is not accurate and  there are restrictions on sharing of data and/or materials, please state these. Please note that we cannot proceed with consideration of your article until this information has been declared.

Additional Editor Comments:

Based on the reviewers' comments and the evaluation of this paper, it is recommended that the paper needs major revision.

Reviewers' comments:

Reviewer's Responses to Questions

**Comments to the Author**

1. Is the manuscript technically sound, and do the data support the conclusions?

Reviewer #1: Yes

Reviewer #2: Yes

2. Has the statistical analysis been performed appropriately and rigorously?

Reviewer #1: N/A

Reviewer #2: N/A

3. Have the authors made all data underlying the findings in their manuscript fully available?

Reviewer #1: Yes

Reviewer #2: Yes

4. Is the manuscript presented in an intelligible fashion and written in standard English?

Reviewer #1: Yes

Reviewer #2: Yes

5. Review Comments to the Author

Reviewer #1: The manuscript presents an investigation into the scalability of displacement-driven compliant mechanisms, addressing an important aspect of compliant mechanism design that has significant implications for multi-scale applications. The authors demonstrate, through experimental validation and detailed case studies, that the strain remains constant across geometric scales while force scales linearly with width, confirming the theoretical foundations proposed.

The discussion appropriately highlights the counterintuitive nature of displacement-driven compliant mechanisms compared to conventional force-driven systems, and, in general, it insightfully points to potential future applications.

Overall, the topic fits the interest of the journal.

Language and writing, under the syntactic and grammar aspects, results in being sufficiently fluent and clean.

The following specific remarks are provided:

1) The manuscript could benefit from a more detailed exploration of potential manufacturing challenges and material limitations that might arise at extreme scales, as well as a clearer explanation of how these scaling principles might integrate with existing design optimization frameworks.

2) The explanation of the theoretical background, while generally sound, lacks sufficient depth and rigor in some parts; for example, the assumptions behind the scaling laws are not always clearly stated or justified, which may leave readers questioning the generality of the results.

3) The discussion section touches on promising applications but tends to be somewhat speculative without clear pathways for practical implementation or addressing potential limitations in manufacturing and material behavior at extreme scales.

4) Furthermore, the paper would benefit from a more critical treatment of challenges related to scaling, such as non-ideal material properties, fabrication tolerances, and environmental factors.

5) The paper also overlooks the possible impact of stress concentrations responses when scaling compliant mechanisms, which could lead to unexpected failure modes (or early failure) during the application of cycles loads.

6) How does fatigue behavior modify when dimensions change? Please, add a discussion on this point in the manuscript.

7) Authors may consider discussing the following reference: in this recent paper on cross-axis flexure hinges (a) https://doi.org/10.1016/j.mechmachtheory.2024.105894, the authors demonstrate that second-order kinematic invariants (specifically the polodes and the inflection circle) scale linearly with the length of the flexure hinge.This finding offers an interesting perspective that could enrich your discussion by highlighting that certain elasto-kinematic properties of compliant mechanisms also exhibit scaling relationships useful for the design.

Downstream of these points, the Reviewer proposes a major revision and defer the publication decision pending the authors' response and modifications according to these remarks.

Reviewer #2: This work shows that for displacement-driven compliant mechanisms, mechanical stress is uniquely invariant with scale. This effect is described theoretically, verified through computer models and physical testing, and is demonstrated in three examples: a parallel-guiding mechanism, a projectile launcher, and a chair.

The work is useful and emphasized scalability challenges and how to overcome them.

Here are comments to improve the work:

- the motivation can be improved on why scalability matters and how to approach it. In the case of compliant mechanisms how scalability can be beneficial for monolithic design

- discuss the actuation of complaint mechanisms as well as the power needed in the context of scalability

- the conclusions need improvement: clearly state the contributions of the work as well as how to interpret the main findings for design

6. PLOS authors have the option to publish the peer review history of their article (what does this mean?). If published, this will include your full peer review and any attached files.

Reviewer #1: No

Reviewer #2: No

---

## [Author Response · Author response to Decision Letter 1]

17 Sep 2025

Revisions 9/17/2025: To the editors: We have added the .tex file for the manuscript to the submission. Note that the abstract does not contain any citations.

Revisions 9/1/2025: Thank you for the information pertaining to journal requirements included in the email. We have made the journal-suggested edits as described in the Response to Reviewers pdf attached to this submission.

---

## [Decision Letter · Decision Letter 1]

16 Oct 2025

PONE-D-25-26746R1 Monolithic scalable compliant mechanisms

PLOS ONE

Dear Dr. Parkinson,

Thank you for submitting your manuscript to PLOS ONE. After careful consideration, we feel that it has merit but does not fully meet PLOS ONE’s publication criteria as it currently stands. Therefore, we invite you to submit a revised version of the manuscript that addresses the points raised during the review process.

After evaluation, please revise the manuscript thoroughly in accordance with the reviewers' comments. Note that the references recommended by the reviewers are not mandatory, unless they are necessary and truly relevant.

We look forward to receiving your revised manuscript.

Kind regards,

Geng Wang, PhD

Academic Editor

PLOS ONE

Journal Requirements:

Additional Editor Comments:

After evaluation, please revise the manuscript thoroughly in accordance with the comments. Note that the references recommended by the reviewers are not mandatory, unless they are necessary and truly relevant.

Reviewers' comments:

Reviewer's Responses to Questions

**Comments to the Author**

1. If the authors have adequately addressed your comments raised in a previous round of review and you feel that this manuscript is now acceptable for publication, you may indicate that here to bypass the “Comments to the Author” section, enter your conflict of interest statement in the “Confidential to Editor” section, and submit your "Accept" recommendation.

Reviewer #3: All comments have been addressed

Reviewer #4: All comments have been addressed

2. Is the manuscript technically sound, and do the data support the conclusions?

Reviewer #3: Yes

Reviewer #4: Yes

3. Has the statistical analysis been performed appropriately and rigorously?

Reviewer #3: Yes

Reviewer #4: N/A

4. Have the authors made all data underlying the findings in their manuscript fully available?

Reviewer #3: Yes

Reviewer #4: Yes

5. Is the manuscript presented in an intelligible fashion and written in standard English?

Reviewer #3: Yes

Reviewer #4: Yes

6. Review Comments to the Author

Reviewer #3: All my indications and/or suggestions have been correctly implemented. The paper is well written and well structured, and the results have been described in a thorough and comprehensive manner. I believe that the paper is ready for publication.

Reviewer #4: This manuscript explores the scalability of displacement-driven compliant mechanisms, demonstrating that mechanical stress remains invariant under geometric scaling. The work combines analytical derivation, finite element analysis, and physical testing across three illustrative case studies: a parallel-guiding mechanism, a projectile launcher, and a deployable chair. The authors argue that this property simplifies design transferability across scales and opens new opportunities in areas such as MEMS, medical devices, and deployable structures.

The revised manuscript addresses most of the issues raised in earlier review rounds, improving clarity, rigor, and relevance. Overall, it represents a valuable and well-supported contribution to the field of compliant mechanism design. The following comments need to be addressed.

1. Add scale bar in Fig. 1, Fig. 3. Also Fig. 1 a, a and c and be laid in single column.

2. Figure 2 sub figures should be labeled as a, b, c and d to improve readability

3. Figure 3 and 4 can be merged to single figure as they are similar and this will make the manuscript more readable.

4. Did author check the projectile launch for all scales? What the distance that the launcher can travel as illustrated in Figure 9. Is MEMS launcher functional as beans are deformed. Can it store the force, it should be supported with video or tracking the displacement.

5. Figure 6 and 7 should be merged as they don’t add more information. How were the sensors on Figure 7 calibrated?

6. A supporting video on formation of chair should be included in the supporting information along with projectile launch.

7. Add scale bar for Figure 10. What weight of the person that the chair can bear without deformation or break.

8. The following statement should be supported by experimental results: ‘A Keyence VHX microscope was used in conjunction with a micromanipulator probe to view and actuate the CICNT devices’

9. Recent work on compliant mechanisms with multi degrees of freedom (DOF) should be included in discussion section. Some examples includes.

i. Yang, H., Patel, D.K., Johnson, T. et al. A compliant metastructure design with reconfigurability up to six degrees of freedom. Nat Commun 16, 719 (2025). https://doi.org/10.1038/s41467-024-55591-2

ii. Humphrey Yang, et al. 2022. ReCompFig: Designing Dynamically Reconfigurable Kinematic Devices Using Compliant Mechanisms and Tensioning Cables. In Proceedings of the 2022 CHI Conference on Human Factors in Computing Systems (CHI '22). Association for Computing Machinery, New York, NY, USA, Article 170, 1–14. https://doi.org/10.1145/3491102.3502065

7. PLOS authors have the option to publish the peer review history of their article (what does this mean?). If published, this will include your full peer review and any attached files.

Reviewer #3: No

Reviewer #4: No

---

## [Author Response · Author response to Decision Letter 2]

5 Nov 2025

We thank the reviewers for their thoughtful comments. Please refer to the 2nd Resolution to Review document for specific changes made.

---

## [Decision Letter · Decision Letter 2]

18 Dec 2025

Monolithic scalable compliant mechanisms

PONE-D-25-26746R2

Dear Dr. Parkinson,

We’re pleased to inform you that your manuscript has been judged scientifically suitable for publication and will be formally accepted for publication once it meets all outstanding technical requirements.

Kind regards,

Geng Wang, PhD

Academic Editor

PLOS One

Additional Editor Comments (optional):

Both reviewers and the editor have no further comments on the revised manuscript and recommend its acceptance.

Reviewers' comments:

Reviewer's Responses to Questions

**Comments to the Author**

1. If the authors have adequately addressed your comments raised in a previous round of review and you feel that this manuscript is now acceptable for publication, you may indicate that here to bypass the “Comments to the Author” section, enter your conflict of interest statement in the “Confidential to Editor” section, and submit your "Accept" recommendation.

Reviewer #2: All comments have been addressed

Reviewer #4: All comments have been addressed

2. Is the manuscript technically sound, and do the data support the conclusions?

Reviewer #2: Yes

Reviewer #4: Yes

3. Has the statistical analysis been performed appropriately and rigorously?

Reviewer #2: Yes

Reviewer #4: Yes

4. Have the authors made all data underlying the findings in their manuscript fully available?

Reviewer #2: Yes

Reviewer #4: Yes

5. Is the manuscript presented in an intelligible fashion and written in standard English?

Reviewer #2: Yes

Reviewer #4: Yes

6. Review Comments to the Author

Reviewer #2: comments have been addressed and the authors have improved the manuscript. The manuscript is in good condition.

Reviewer #4: Authors have addressed the comments and incorp[orated in the revised manuscript. Best wishes to the authors.

7. PLOS authors have the option to publish the peer review history of their article (what does this mean?). If published, this will include your full peer review and any attached files.

Reviewer #2: No

Reviewer #4: No

---

## [Editor Report · Acceptance letter]

PONE-D-25-26746R2

PLOS One

Dear Dr. Parkinson,

I'm pleased to inform you that your manuscript has been deemed suitable for publication in PLOS One. Congratulations! Your manuscript is now being handed over to our production team.

Kind regards,

on behalf of

Professor Geng Wang

Academic Editor

PLOS One